# Can we reliably reconstruct the mid-Pliocene Warm Period with sparse data and uncertain models?

James D Annan[1], Julia C Hargreaves[1], Thorsten Mauritsen[2], Erin McClymont[3], and Sze Ling Ho[4]

[1]Blue Skies Research Ltd, The Old Chapel, Albert Hill, Settle, BD24 9HE, UK
[2]Department of Meteorology, Stockholm University, Stockholm, Sweden
[3]Department of Geography, Durham University, Durham, DH1 3LE, UK
[4]Institute of Oceanography, National Taiwan University, 10617 Taipei, Taiwan

*Correspondence to:* jdannan@blueskiesresearch.org.uk

**Abstract.**

We present a reconstruction of the surface climate of the mid-Pliocene Warm Period (mPWP), specifically Marine Isotope Stage (MIS) KM5c or 3.205 Ma. We combine the ensemble of climate model simulations which contributed to the PlioMIP projects, with compilations of proxy data analyses of sea surface temperature (SST). The different data sets we considered are all sparse with high uncertainty, and the best estimate annual global mean surface air temperature (SAT) anomaly varies from 1.0 up to 4.7°C depending on data source.

We argue that the latest PlioVAR analysis of alkenone data is likely more reliable than other data sets we consider, and using this data set yields a SAT anomaly of $3.6 \pm 1.0$°C, with a value of $2.8 \pm 0.9$°C for SST (all uncertainties are quoted at one standard deviation). However, depending on the application, it may be advisable to consider the broader range arising from the various data sets, to account for structural uncertainty. The regional scale information in the reconstruction may not be reliable as it is largely based on the patterns simulated by the models. Nevertheless, despite the uncertainties, there is some indication that polar amplification may be underestimated in the models.

## 1 Introduction

The mid-Pliocene Warm Period (mPWP, also mid-Piacenzian Warm Period), roughly 3.2 Ma, represents the most recent period when atmospheric $CO_2$ was significantly higher than the pre-industrial level for a sustained period, and as such is an attractive target for analysis in order to better understand how our modern climate system may respond to sustained elevated greenhouse gas concentrations. It has been the focus of two international climate modelling projects, PlioMIP1 (Haywood et al., 2010) and PlioMIP2 (Haywood et al., 2016) within each of which a common protocol was developed in order that multiple climate models from different research centres could generate simulations which may be compared to each other and also to proxy data collected and analysed from this time interval.

While models provide global fields that attempt to represent the climate state at that time, these outputs do not directly use surface temperature information from proxy data and their representation of the physics of the climate system is imperfect. There have been few attempts to estimate the climate state (e.g. global temperature fields) based on proxy data, and these have

generally used statistical approaches which do not take advantage of the physical constraints represented by climate models. For example, Bragg (2014) generated a surface temperature reconstruction based on a Gaussian Markov random field using the earlier PRISM3 data, but this method does not account for physical properties of the climate system such as land-sea contrast or polar amplification which are underpinned by physical theory and robustly reproduced in models. When data are sparse and uncertain, as is the case here, it is necessary to interpolate to unobserved regions. Inglis et al. (2020) compared numerous heuristic and statistical methods to generate global mean temperature estimates for periods considered in the DeepMIP project (which did not include the mPWP). Data assimilation methods which combine the physics embedded in climate models, with the sparse and uncertain observations that are available, have the potential to improve on methods that rely on either models or data alone.

In Annan and Hargreaves (2013), we presented a novel approach to reconstructing the surface temperature fields (both surface air temperature, or SAT, and sea surface temperature, or SST) for the Last Glacial Maximum (LGM, 19–23ka) which combined the "ensemble of opportunity" of model simulations of this time period together with compilations of proxy-based surface temperature estimates. The inputs and methods were updated in Annan et al. (2022) (henceforth AHM22). Here we apply the same method as presented in AHM22 to reconstruct the surface temperature fields of the mPWP, specifically MIS KM5c or 3.205 Ma. We combine the ensemble of climate model simulations which contributed to the PlioMIP projects, together with compilations of SST proxy data, to provide an estimate of the climate state at the mPWP. We introduce the model ensemble in Section 2. Two compilations of proxy data relating to this period have been published by Foley and Dowsett (2019) and McClymont et al. (2020), which we discuss further in Section 3. We discuss methodological choices and their impact in Section 4 and summarise our results in Section 5.

## 2 Models

There have been two major modelling projects focusing on the mPWP. The first Pliocene Modelling Intercomparison Project (Haywood et al., 2010), which we refer to as PlioMIP1 for clarity, comprised 9 models simulating a generic interglacial period in the range of 3.264 to 3.025 Ma BP. For the second Pliocene Modelling Intercomparison Project or PlioMIP2 (Haywood et al., 2016), the specific interglacial period MIS KM5c was simulated, dated at 3.205 Ma BP. The boundary conditions thus varied slightly between the two projects, but the importance of these differences in determining the outputs of the simulations is small compared to the differences in physics between the models. Thus as our initial ensemble, we take the results of all the model simulations from both projects.

There are 9 simulations available from PlioMIP1 and 16 from PlioMIP2, with the models being listed in Table 1. In order to perform our synthesis, model outputs were interpolated onto a regular $2 \times 2°$ grid for SAT and a $5 \times 5°$ grid for SST. The ocean boundary varies slightly between models, due to various differences in resolution and underlying model grid. Our algorithm requires all ensemble members to generate output at each grid point, and thus we mask out all locations on the SST grid where any model has land. While this has the unfortunate effect of reducing the number of usable data points, we note that the points

| Experiment | Model Name | GSAT °C |
|---|---|---|
| PlioMIP1 | (AWI-COSMOS-1) | 3.6 |
| PlioMIP1 | CCSM4-1 | 1.8 |
| PlioMIP1 | FGOALS | 4.2 |
| PlioMIP1 | GISS-1 | 2.2 |
| PlioMIP1 | (HADCM3-1) | 3.3 |
| PlioMIP1 | (IPSL-1) | 2.0 |
| PlioMIP1 | (MIROC4-1) | 3.5 |
| PlioMIP1 | MRI-1 | 1.8 |
| PlioMIP1 | NorESM-1 | 3.2 |
| PlioMIP2 | AWI-COSMOS-2 | 3.4 |
| PlioMIP2 | CCSM4-2 | 2.7 |
| PlioMIP2 | CESM1.2 | 4.0 |
| PlioMIP2 | CESM2.1 | 3.1 |
| PlioMIP2 | EC-Earth3.3 | 4.9 |
| PlioMIP2 | GISS-2 | 2.1 |
| PlioMIP2 | HadCM3-2 | 2.9 |
| PlioMIP2 | HadGEM3 | 5.1 |
| PlioMIP2 | IPSLCM5A | 2.3 |
| PlioMIP2 | (IPSLCM5A2) | 2.2 |
| PlioMIP2 | MIROC4m-2 | 3.1 |
| PlioMIP2 | MRI-CGCM2-3 | 2.4 |
| PlioMIP2 | NorESM-L-2 | 2.1 |
| PlioMIP2 | NorESM1-F-2 | 1.7 |
| PlioMIP2 | UofT-CCSM4 | 3.8 |
| PlioMIP2 | Utrecht-CCSM4 | 4.7 |

**Table 1.** Models available for the mPWP reconstruction, and their global mean surface air temperature anomalies for the mPWP. Parenthesis indicate models which were removed due to similarity. See text for details.

that are masked in this way are coastal in location, which are potentially problematic for data-model comparisons due to the local nature of upwelling dynamics that is not always adequately captured by models.

As Table 1 indicates, some models either directly contributed more than one simulation to the combined ensemble, or else were closely related to other models which participated. Therefore we performed a thinning process similar to that described in AHM22. The goal is to generate an ensemble where we do not expect to find any clustering where particular sets of models have highly similar outputs. We examined the literature to consider which models were closely related through their construction

to other models in the ensemble. Then we checked this assessment by calculating RMS differences and correlations between the simulated anomalies. We concluded that the following subsets of models were too similar for them to contribute more than one member to our ensemble: (AWI-COSMOS-1, AWI-COSMOS-2), (HADCM3-1, HadCM3-2), (IPSL-1, IPSLCM5A1, IPSLCM5A2), (MIROC4-1, MIROC4m-2). From these, we removed the models as indicated in the table, retaining the newest
of each set, leaving us with a final ensemble of 20 model simulations. The set of global mean SAT values from this ensemble $(3.1 \pm 1.1°C$ at one standard deviation) passed a Shapiro-Wilk test of normality.

The ensemble mean surface temperature anomaly fields for this 20-member ensemble are shown in Figure 1, along with other results that we will discuss later. The extent of our land mask is shown in the SST plots. The location and values of one set of data points are also indicated on the SST field, and we discuss these in the following section.

## 3 Proxy data

We consider various data sets in this paper, all of which have been calibrated to SST. The most widely used data sets for analysis and interpretation of the mPWP have been those generated by various iterations of the PRISM projects (Foley and Dowsett, 2019). Here we primarily consider the PRISM4 proxy estimates for SST from a 30ka-wide interval centred on MIS KM5c (Foley and Dowsett, 2019), that is $3.205 \pm 0.015$ Ma. We also consider their SST estimates based on data limited to
15 the more restricted interval of $3.205 \pm 0.005$ Ma, but the differences between these two analyses are very minor. PRISM4 SSTs were compiled from published literature, using an alkenone SST proxy (the $U_{37}^{K'}$ index) which is converted to mean annual SST using a linear calibration based on globally distributed surface sediment data (Müller et al., 1998). Anomalies from the pre-industrial climate are calculated relative to the NOAA ERSSTv5 data set (Huang et al., 2017). This PRISM4 compilation contains 37 data points, reducing to 34 distinct grid points on the regular $5 \times 5$ degree grid that we use for our
SST analysis, of which 23 locations remain after masking to the ocean grid of our ensemble. The tighter $3.205 \pm 0.005$ Ma compilation of PRISM4 data contained only 33 points, but these still covered the same 23 grid points on the ocean grid of our ensemble. No uncertainty estimates were included with these data. We assume a value of $2°C$ for the uncertainty at one standard deviation, which is larger than the value of $1°C$ that we used for various proxy estimates when reconstructing the Last Glacial Maximum (Annan and Hargreaves, 2013), but may still be optimistic, given there is generally less certainty for older
paleo-periods. We discuss this choice in Section 4.

More recently, the PlioVAR project (McClymont et al., 2020) produced an updated analysis of proxy data to examine the 20ka-wide interval of $3.205 \pm 0.01$ Ma. The PlioVAR project required data to have a minimum temporal resolution ($\leq$ 10 kyr) and be constrained in time by an orbital-scale age model. Several PRISM4 sites were excluded as a result of this strict stratigraphic protocol, and for others there was a revision to the original published age model where the MIS KM5c peak had
30 been misaligned (McClymont et al., 2020). Within the PlioVAR data set we consider separately the $U_{37}^{K'}$ data points, of which there are 23, and the Mg/Ca data points of which there are 13.

For the $U_{37}^{K'}$ data, two SST calibrations were available and compared by PlioVAR: the linear Müller et al. (1998) calibration, and a more recent Bayesian calibration (Tierney and Tingley, 2018). At high and mid latitudes the reconstructed SSTs differ by

less than 1°C, whereas in the equatorial region the difference may be up to 1.5°C (McClymont et al., 2020), which are smaller than the individual calibration errors. The $U_{37}^{K'}$ SST values we take here are the simple average of the calibration of Müller et al. (1998), and the BAYSPLINE calculation, as presented in McClymont et al. (2020).

In McClymont et al. (2020), the Mg/Ca SST values presented were unaltered from the original estimates in the source publications. Subsequently, the calibrations were revisited and consistent corrections applied including for evolving seawater Mg/Ca and potential dissolution (McClymont et al., 2023). This updated data set also includes one data point based on the organic proxy "TEX$_{86}$" (Schouten et al., 2002), which we include within the Mg/Ca dataset for convenience. For Mg/Ca, we use the newer values where available, and the calibration presented in McClymont et al. (2020) for the remainder. We discard the Mg/Ca SST estimates generated by a Bayesian calibration (BAYMAG) which were also presented in McClymont et al. (2020), although we have checked that including these does not qualitatively change our analysis, as we describe further below. The data values that we use here are contained in the supplementary information in order to ensure reproducibility.

In cases where multiple data points from the same proxy type coincide on the grid, we use simple averages. In the case where both Mg/Ca and $U_{37}^{K'}$ data coincide in the same box, we then average their averages, thereby giving equal weight to the Mg/Ca and the $U_{37}^{K'}$. We refer to the data sets thus derived as PlioVAR-$U_{37}^{K'}$, PlioVAR-Mg/Ca and PlioVAR-all. The locations and values of the data sets used can be seen in the SST plots of Figure 1. As for the PRISM4 data, uncertainties of 2°C are assumed for the final gridded and averaged values.

The PRISM4 and PlioVAR data sets overlap substantially in locations. This is unsurprising, since they are based on largely the same set of drilled cores. There are 17 grid points in common between PRISM4 and PlioVAR-$U_{37}^{K'}$, and the values at these locations are highly consistent, matching well with a mean difference of 0.0°C and an RMS difference of 0.7°C. The correlation of the two data sets at these coincident locations is also very high at 0.97. While the new PlioVAR chronology and recalibration have introduced changes in SST values at the grid-point level, these have broadly cancelled out overall. McClymont et al. (2020) highlighted some inconsistencies between the Mg/Ca data and both the $U_{37}^{K'}$ data and the PlioMIP models. There are 6 locations in common between the PRISM4 and PlioVAR-Mg/Ca data sets and these also have a very substantial mean difference of 3.3°C (PlioVAR-Mg/Ca being cooler) with a residual RMS of 5.1°C. It does not seem possible that both data sets are faithful representations of the annual mean sea surface temperature that is required for our analysis. As discussed extensively in previous studies (e.g. De Schepper et al., 2013; Dekens et al., 2008; Evans et al., 2016; Karas et al., 2020; Lawrence and Woodard, 2017; Leduc et al., 2014; Naafs et al., 2020), the relationship between Mg/Ca and annual mean SST may depend on a range of site-specific factors including the species analysed, the calibration used, or the seasonality or depth habitat of the foraminifera. Therefore, since our focus here is on the determination of the annual surface temperature signal recorded by the proxies, we consider the Mg/Ca ratio to have a more complex relationship to SST than the $U_{37}^{K'}$ data (e.g. McClymont et al., 2023)

The Mg/Ca might be considered less reliable as their relationship to SST may depend on site-specific factors including the species analysed, the calibration used, or the seasonality or depth habitat of the foraminifera.

Of the three data sets, PlioVAR-$U_{37}^{K'}$ shows the closest agreement with the models, with a pointwise RMS difference of 2.2° C from the ensemble mean, whereas the PlioVAR-Mg/Ca and PRISM4 data sets differ from the ensemble mean by almost

| Data set | # points | # gridded points | $\Delta T$ | usable points | $\Delta T$ | GSAT (GSST) °C | Polar Amp. |
|---|---|---|---|---|---|---|---|
| PRISM4 | 37 | 34 | 4.0 | 23 | 3.7 | $4.7 \pm 1.0$ ($3.3 \pm 0.9$) | 2.9 |
| PlioVAR-$U_{37}^{K'}$ | 23 | 22 | 4.1 | 14 | 3.6 | $3.6 \pm 1.0$ ($2.8 \pm 0.9$) | 2.5 |
| PlioVAR-Mg/Ca | 13 | 13 | 0.1 | 12 | 0.2 | $1.0 \pm 1.0$ ($1.0 \pm 0.9$) | 12.4 |
| PlioVAR-all | 33 | 32 | 2.7 | 23 | 1.9 | $2.3 \pm 1.0$ ($2.2 \pm 0.9$) | 3.4 |

**Table 2.** Summary of data sets and main results. Columns are: number of points in the data set; number of points in the 5 degree grid; average temperature of the data points; annual mean global air (sea) temperature from the climate reconstruction using the data set; polar amplification of the reconstruction. All uncertainties quoted at one standard deviation.

3° C and 4°C respectively. This latter figure may seem surprising since the PlioVAR-$U_{37}^{K'}$ and PRISM4 data agree rather well where both data sets coincide, but apparently the PRISM4 cores that are not included in the PlioVAR-$U_{37}^{K'}$ data set show a markedly greater divergence from the models.

Our preferred reconstruction is thus based on the PlioVAR-$U_{37}^{K'}$ data set, but we also calculate reconstructions using the
5 other data sets to explore the low level of agreement between proxies, and it may be advisable to use the full range of results we present in order to account for the uncertainties in the proxy data analysis. The locations and values of the alternative SST data sets used can be seen in the SST plots of Figure 4. All four data sets, and the climate reconstructions they generate, are summarised in Table 2.

While there are some proxy data relating to the land surface (Salzmann et al., 2008, 2013), the dating and precision in terms
of inferred temperature do not yet appear to be sufficiently robust for us to use here. Multiple proxies covering land and ocean would enable more precise and robust climate reconstructions and we expect that the analysis presented here may be improved in future years.

## 4   Method

The method follows that of AHM22. That is, we firstly perform a recentering step, to shift the ensemble mean towards the data. This is done through a pattern scaling approach in which we fit the first 4 empirical orthogonal functions (EOFs) of the ensemble to the data points, and take this result as our ensemble mean. Our justification for this step, as in AHM22, is that we do not consider the PlioMIP ensemble of opportunity to be a reliable prior, since the models may share common biases in both their physics and the boundary conditions that are applied. By this re-centering procedure, AHM22 demonstrated that we can
strongly reduce the influence of possible biases in the model ensemble on the posterior. Some evidence for this is shown in the rank histograms of Figure 2, the left panel of which compares the data points to the the original PlioMIP ensemble. With so few data points, the evidence of a bias between models and data is not compelling but a large majority of points lie in the right hand side of the rank histogram with 6 of the 14 points in the top quarter. Recentering the prior leads to a more even distribution of data points across the ensemble as shown in the right panel of Figure 2. Ideally, the rank histogram should be

flat with a uniform spread of data points. The recentering step does not alter the covariance structure of the ensemble, merely shifting the mean. We then perform a standard Ensemble Kalman Filter (EnKF) assimilation step using the proxy data, in a similar manner that of AHM22 and Tierney et al. (2020).

Unlike the reconstruction of the Last Glacial Maximum described in AHM22, however, the recentering step here has a striking influence on the final result. This is simply because the data lie some way from the ensemble mean, being generally warmer, as the rank histograms indicate. If we believed that our model ensemble was a good prior, then it would be appropriate to omit the recentering step and perform the assimilation directly on the PlioMIP ensemble. However, the EnKF assimilation only updates the ensemble in the neighbourhood of data points (due to both the generally local nature of the empirical correlations in the ensemble, and the imposed localisation) and therefore cannot interpolate far into data-void regions. Thus, our final result would be highly dependent on the model prior. To illustrate the effect of our methodological choices, here we compare our two step results with this alternative framework. Figure 3 compares our two-step process with the single step EnKF approach using the PlioMIP models as a prior without the recentering step. The top two pairs of panels show the magnitude of change in the ensemble mean, first from centering the ensemble on the data (via the EOF pattern scaling as described in AHM22), and then from applying the EnKF algorithm to this ensemble. Note the factor of ten change in scaling on the colour bars between the plots of the recentering and the EnKF steps. The EnKF update generates changes only in the vicinity of the data points, due both to the localisation which was imposed (a length parameter of 2500km in the Gaspari and Cohn (1999) scheme was used here, as in AHM22, corresponding to a cut-off radius of 5000km) and also the typically local nature of covariances across the ensemble. The changes are small even at the data locations, due to the low density of data points and the large uncertainties assumed for these proxy data. If we were to assume a lower uncertainty on these points, then the update would be somewhat larger but would still be limited to the neighbourhood of the data points.

If we do not perform the recentering and instead just perform the single step standard EnKF update to the original PlioMIP ensemble, the update is slightly greater than the EnKF step in the two-step method, and generally more positive, due to the data being warmer than the ensemble mean. However the increment is still small, as the data are sparse and uncertain. Large areas of the globe are almost unaffected by the assimilation, with a temperature change of less than 0.1°C. This is an inevitable consequence of having limited sparse data, and points to the influence of the model prior on the final result. Thus, with so few and uncertain data points the final result using a one-step framework would be very heavily based on the initial ensemble. This result is very different from that of AHM22, where the recentering step had relatively small influence on the final result and the much larger data set of around 400 data points, scattered across both land and ocean, meant that the EnKF update step was more substantial. When the recentering step is omitted, data cannot possibly influence the posterior beyond the cut-off radius, and therefore we also tested the analysis with larger length parameters of up to 10,000km (corresponding to a cut-off radius of 20,000km). However our results were unchanged to within rounding error, demonstrating that the choice of length scale was not a significant factor in our results.

We take a uniform uncertainty of 2°C on all of our data points. With so few data points, this estimate is necessarily itself uncertain, but we consider it reasonable for the following arguments. The RMS difference between the original PlioMIP ensemble members and the data points is around 2.6°C averaged across the ensemble members, or 2.2°C relative to the

ensemble mean, which precludes a much higher error value since the data should not be closer to the models than they are to reality under the assumption that model errors and data errors are independent. Conversely, our posterior mean estimate after fitting to the data only achieves a residual RMS difference of 1.7°C. While we could hope that the data uncertainties are smaller than we have achieved, we note that the RMS difference between the PRISM4 data and the Mg/Ca data is greater than 5°C (at the 6 coincident points) which is a worrying indication that the calibration to SST may not be adequately understood at this point. In Section 3 we argued that the Mg/Ca may be less reliable. We do not explore possible reasons for this here, but they may include site-specific factors including the species analysed, the calibration used, or the seasonality or depth habitat of the foraminifera (e.g. McClymont et al., 2023, and references therein).

We adopt the same parameters for the algorithm that were shown to work well in AHM22, of 4 EOFs, and a localisation length parameter of 2500km implying a cut-off radius of 5000km. While changing these values altered the regional patterns somewhat (for example, using a larger number of EOFs introduced more spatial variability) they made little difference to the large scale results such as global mean temperature anomalies. In AHM22, we performed extensive validation of the method, and the choice of parameters, through a process of leave-one-out cross validation. Here, however, we have so few data points that this process cannot be useful, as the sampling error on the statistics will be much larger than any realistic estimate of the underlying effects being investigated.

## 5 Results

### 5.1 Mean state

We create climate state reconstructions using the filtered PlioMIP model ensemble described in section 2, together with four different data sets. The data sets we use are PRISM4, PlioVAR-$U_{37}^{K'}$, PlioVAR-Mg/Ca, and PlioVAR-all, as described in Section 3. The global mean temperature estimates obtained are shown in Table 2.

We find that the PRISM4 and PlioVAR-$U_{37}^{K'}$ data sets both generate reconstructions towards the high end of previous estimates, but not inconsistent with them (e.g. the value of $3 \pm 1$°C assessed by Sherwood et al., 2020). As the table shows, there are differences between the global fields generated from PRISM4 and PlioVAR-$U_{37}^{K'}$ data even though the data points in these two sets take broadly similar values where they exist in both sets of data. In contrast, the PlioVAR-Mg/Ca data generates a strikingly cooler result, and the inclusion of this Mg/Ca data in PlioVAR-all generates a result with considerably less warming than for PRISM4 and PlioVAR-UK37. The reason for these results is simply that the Mg/Ca data themselves indicate very little warming, and even a cooling in the tropics (McClymont et al., 2020).

Due to the updated chronology and more robust calibration using two approaches, we expect that the PlioVAR-$U_{37}^{K'}$ data are likely to be more reliable than PRISM4 and therefore we take as our primary reconstruction the PlioVAR-$U_{37}^{K'}$ result with global annually averaged SAT (SST) of $3.6 \pm 1.0$ $(2.8 \pm 0.9)$°C. In comparison, the original 20-member PlioMIP ensemble (that is, after deletion of near-duplicates) has a global mean SAT anomaly distribution of $3.1 \pm 1.1$°C. The middle plots in Figure 1 show the fields obtained by using the PlioVAR-$U_{37}^{K'}$ data, while the lower plots on the same figure show the difference between

these results and the original PlioMIP ensemble mean. For completeness, the SAT and SST fields obtained from the other three data sets are shown in Figure 4.

## 5.2 Polar Amplification

Following Haywood et al. (2020), we use as our definition of polar amplification the ratio of warming poleward of 60° in both hemispheres, to the global mean change (Smith et al., 2019). The polar amplification in our combined PlioMIP ensemble mean is 2.3, the same as that found by Haywood et al. (2020), but increases markedly to 2.9 in the posterior when using PRISM4 and less substantially to 2.5 when using PlioVAR-$U_{37}^{K'}$. Much larger polar amplification is obtained when the Mg/Ca data are included, due to the greatly reduced tropical warming. While we have low confidence in regional results due to the limited and imprecise nature of the data, these results do hint that the models may underestimate the degree of polar amplification exhibited by the climate system.

## 6 Discussion and Conclusion

We have made the first combined air and sea temperature field reconstructions for the mid-Pliocene Warm Period, using various available data sets and a 20 member model ensemble. For the data set in which we have most confidence (PlioVAR-$U_{37}^{K'}$), our result for annual average global mean surface air (sea) temperatures is $3.6 \pm 1.0$ ($2.8 \pm 0.9$) K. However, the different data sets produce rather different estimates ranging from 1.0 to 4.7 °C for the best estimate of global surface air temperature anomaly. All the data sets are sparse with high uncertainty, and therefore our confidence in our result is not very high. We think that the regional scale information in the reconstruction is not likely to be reliable as it mostly arises during the re-centering procedure, which simply rescales the model patterns including in regions where there are no data. A larger data set would improve this. There is, however, some indication that polar amplification is underestimated in the models.

With such small data sets as we have here, the models also necessarily play an uncomfortably large role. In all of our experiments, the assimilation step of the ensemble Kalman filter makes a relatively small change to the prior. We have investigated the effect of using the original PlioMIP models themselves as a prior, versus recentering the ensemble on the data. This choice has a much more significant influence on the results. In principle we prefer the data-centred approach, because the models could easily be biased through poorly modelled physical processes or uncertainties in boundary conditions, but this is not an unquestionable choice to make. The issue is simply that the data typically available for reconstructions of paleoclimates are insufficient to overcome prior differences.

Even for the LGM, where there is an order of magnitude more proxy data than we have for the mPWP, substantial differences arise between reconstructions that are based on different climate models (e.g. Tierney et al., 2020; Annan et al., 2022). Recently, Masoum et al. (2024) demonstrated how merely switching between two different glacial forcing priors could generate substantially different reconstructions of the last deglaciation, in a data assimilation system that otherwise remained wholly unchanged. We therefore suggest that the use of a single climate model to generate a prior in reconstructions of paleoclimates cannot reasonably be considered robust until validated through the use of different climate models. While our results using

the multi-model ensemble could of course still be erroneous, it at least includes an element of structural uncertainty in their different representations of the climate system.

It is also striking in our results that different data sets show such large discrepancies, especially the disagreement between the Mg/Ca data of McClymont et al. (2020) and the PRISM4 data described in Foley and Dowsett (2019). It seems implausible that both of these data sets could have been reliably calibrated to the annual mean SST anomaly that our algorithm requires, and we suggest that further research is necessary to understand this disagreement.

## 7    Author contributions

JDA, JCH and TM conceived the idea for this study, whereas JDA and JCH performed the analyses and wrote the manuscript. ELM and SLH compiled and reviewed the SST data after synthesis by the PlioVAR working group, discussed data calibrations and uncertainties, and contributed to the writing of the text and edits/discussion with the lead authors.

## 8    Competing interests

One author (ELM) is a member of the Editorial Board of Climate of the Past

## 9    Acknowledgements

This project was funded by the European Research Council (ERC) (Grant agreement No.770765) and the European Union's Horizon 2020 research and innovation program (Grant agreements No.820829 and No.101003470).

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

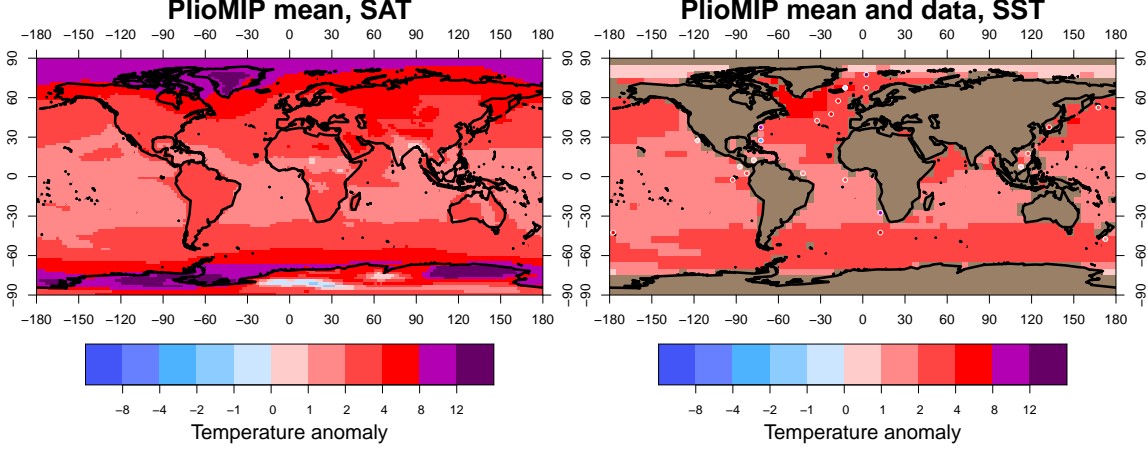

(a) PlioMIP ensemble means, SAT and SST (with proxy data).

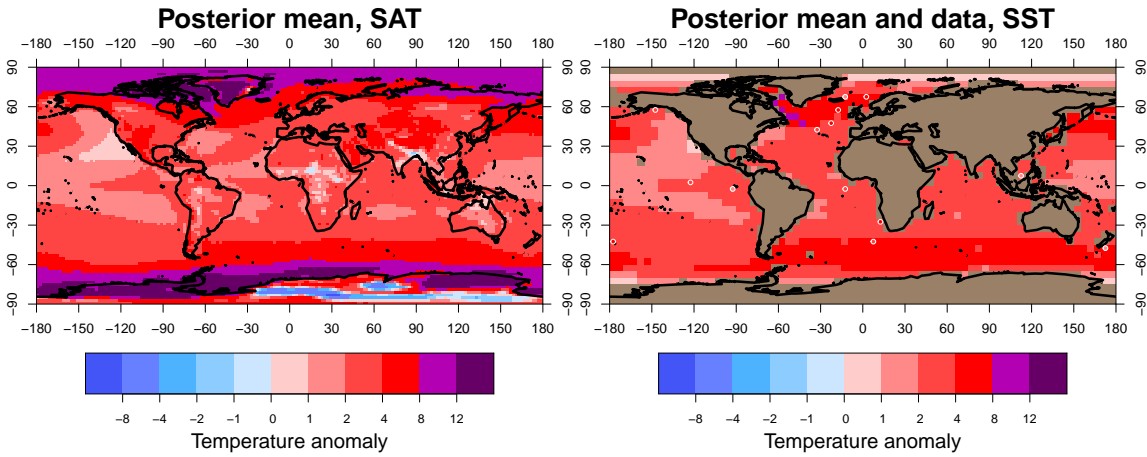

(b) Posterior means, SAT and SST (with proxy data).

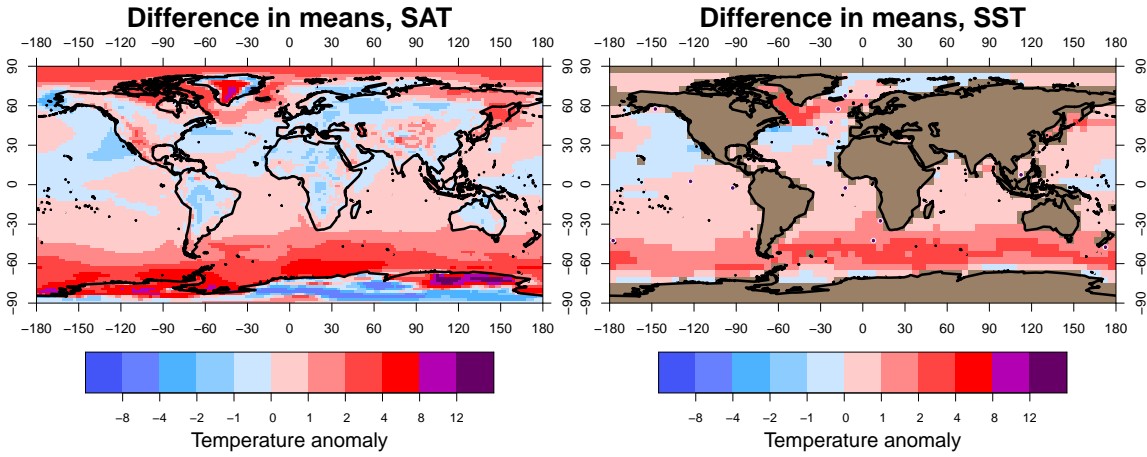

(c) Difference between final reconstruction and PlioMIP ensemble means (location of proxy data points shown).

**Figure 1.** Initial PlioMIP ensemble mean and results obtained using PlioVAR-$U_{37}^{K'}$ data set.

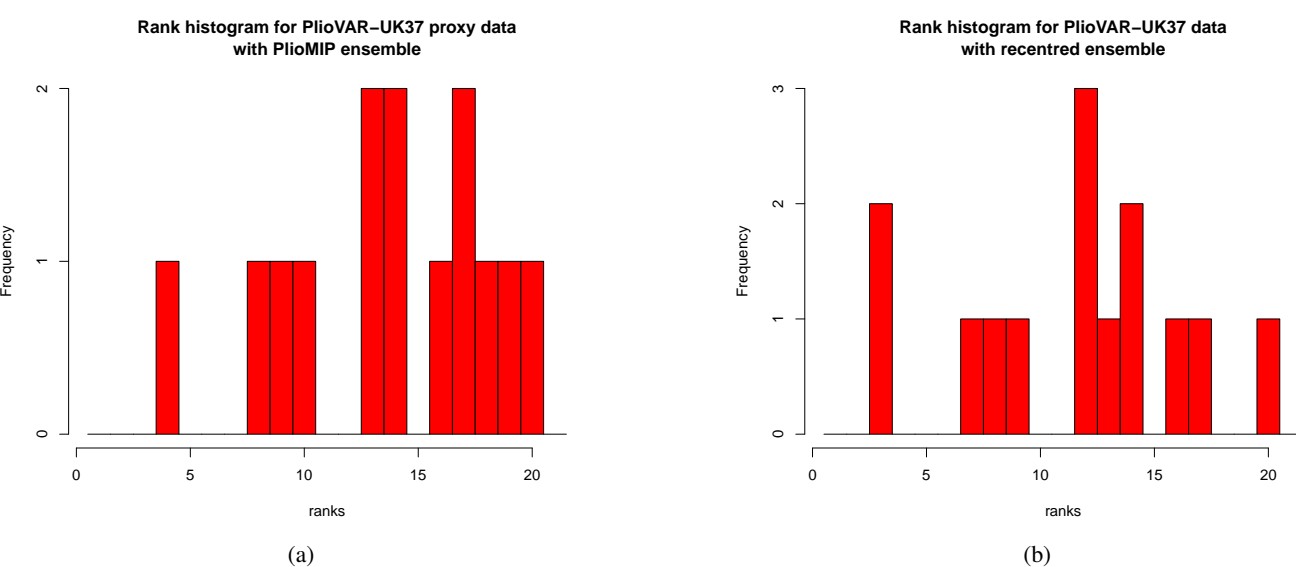

**Figure 2.** Rank histograms of PlioVAR-$U_{37}^{K'}$ data (a) in original PlioMIP ensemble and (b) recentred ensemble.

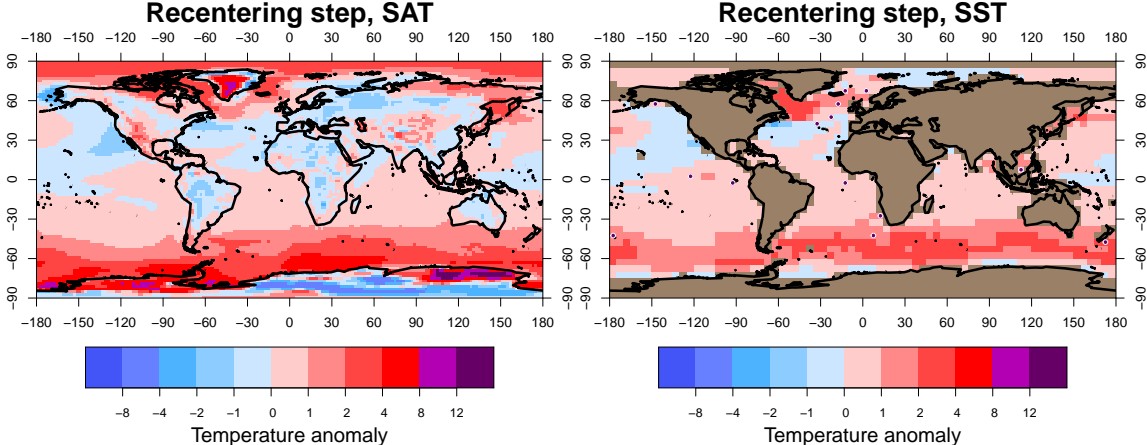

(a) Recentering step of two-step process

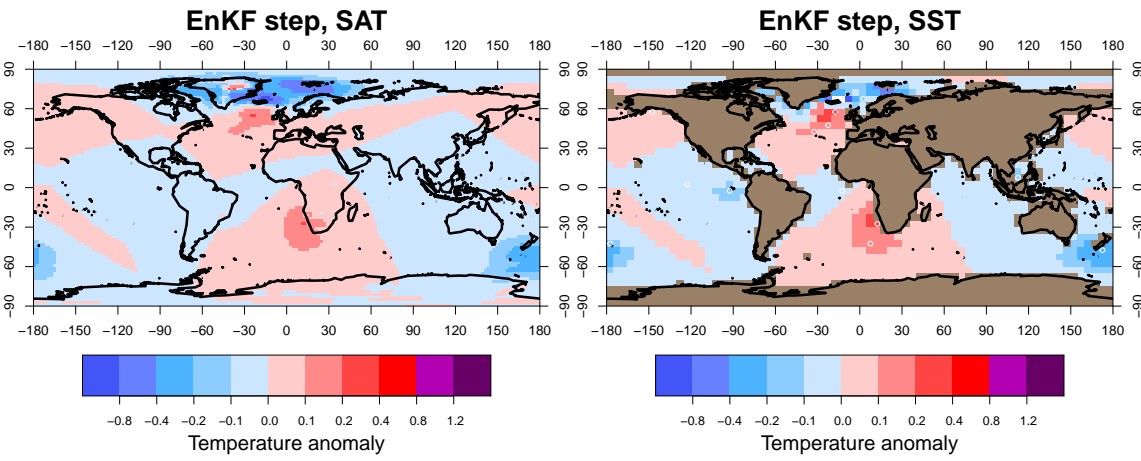

(b) EnKF step of two-step process.

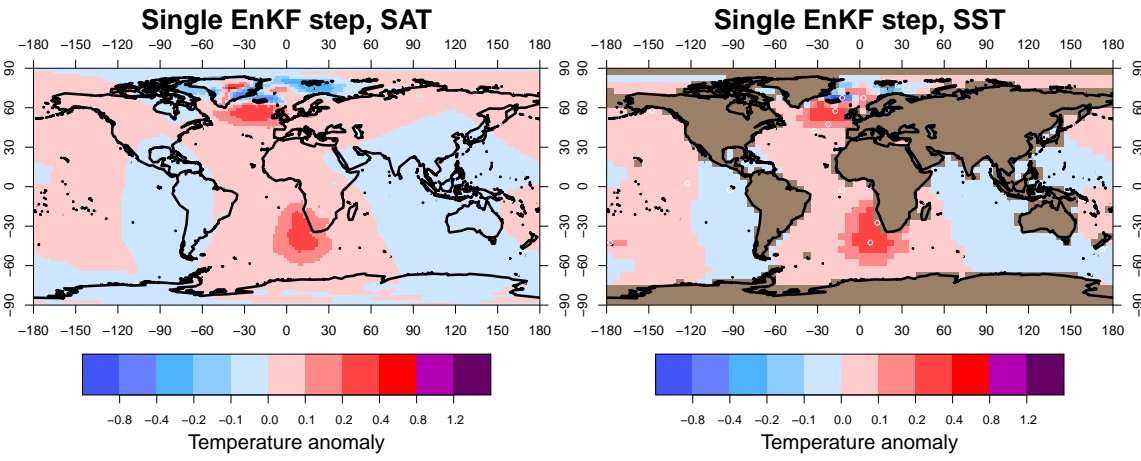

(c) EnKF step of one-step process.

**Figure 3.** Comparison of two-step (recentering and Enkf) versus single step (EnKF only) process. Figures (a) and (b) show the increments generated in the recentering step and EnKF step respectively in the two-step process using PlioVAR-$U_{37}^{K'}$. Figure (c) shows the increment generated in a single step EnKF procedure. Note the different scale on the colour bar for the EnKF steps

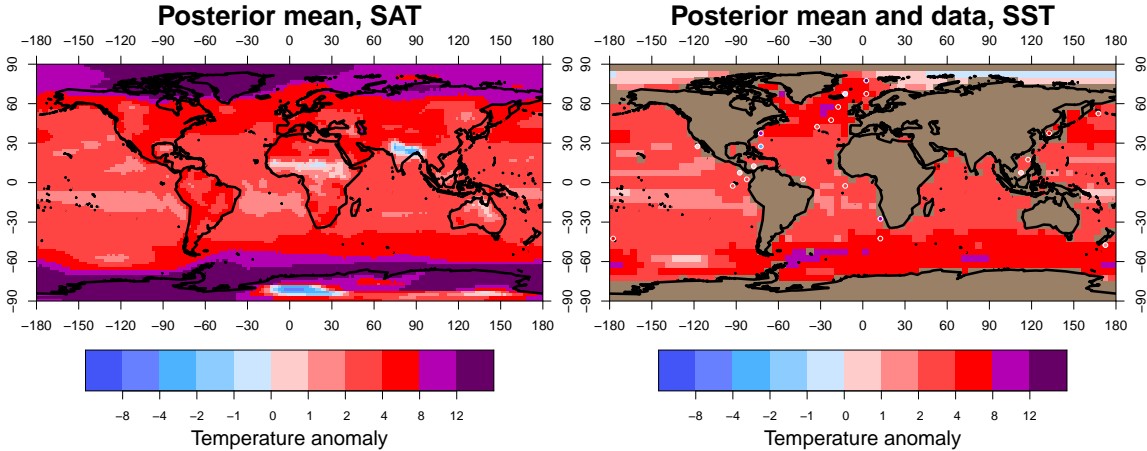

(a) Posterior means using PRISM4 data

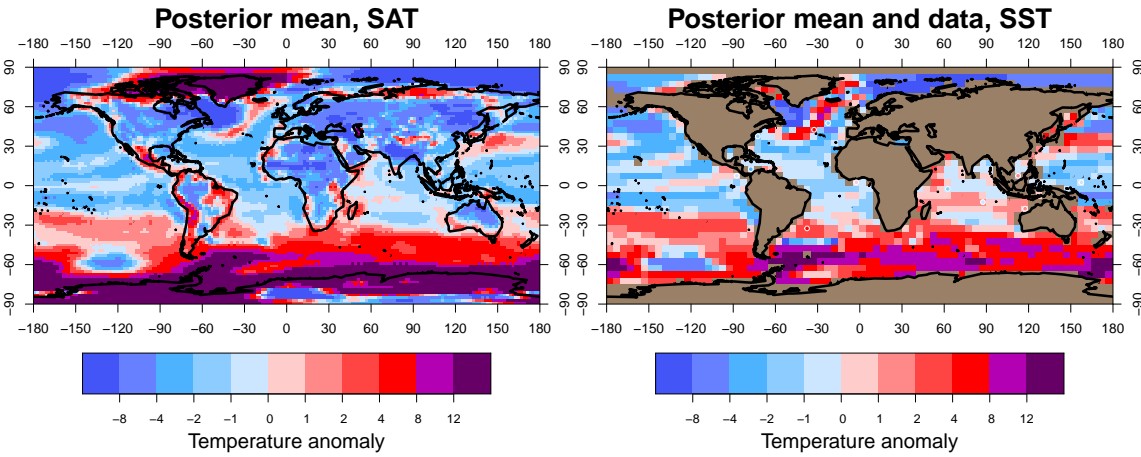

(b) Posterior means, using PlioVar-Mg/Ca data.

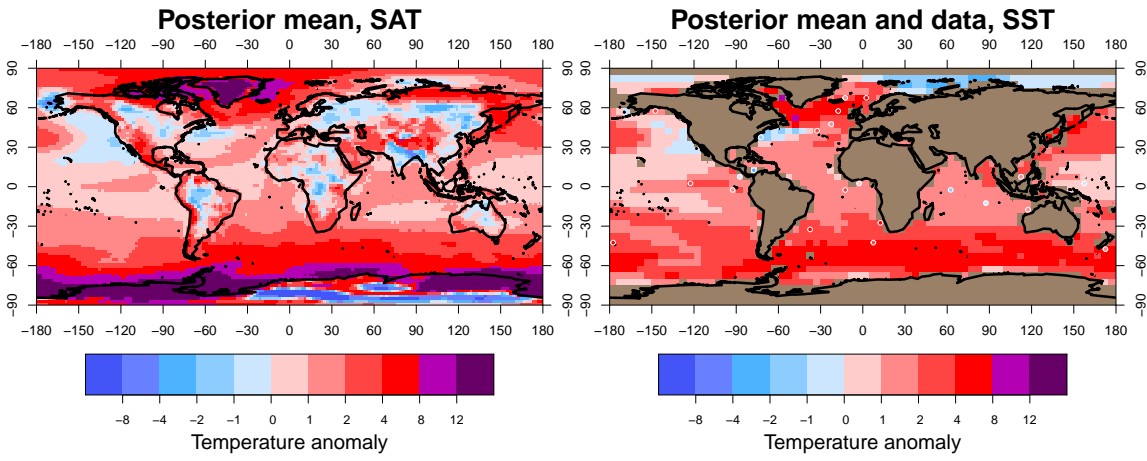

(c) Posterior means, using PlioVar-all data.

**Figure 4.** Results using three different data sets.

