# Peer review of "Can we reliably reconstruct the mid-Pliocene Warm Period with sparse data and uncertain models?"

_EGUsphere, 2023_

## Referee Comment (RC2)

**Can we reliably reconstruct the mid-Pliocene Warm Period with sparse data and uncertain models?**

Annan et al. *submitted to Climate of the Past* (https://doi.org/10.5194/egusphere-2023-1941)

In this contribution, Annan et al. attempt to reconstruct mid-Pliocene Warm Period (mPWP) surface temperatures by combining an ensemble of PMIP-based model simulations with prior compilations of mPWP sediment proxies (namely, alkenones and Mg/Ca). They report a likely mean global atmospheric surface warming of ~3.6 ± 1°C (for their "preferred" solution) relative to the pre-industrial state, with estimates ranging anywhere from 1.0 to 4.7°C average warming depending on the particulars of their chosen proxies (Table 2).

The analysis is interesting, insofar as it represents the first (that I'm aware) attempt at assimilating the mPWP. However, numbers like those mentioned above matter — they get passed on in the literature for generating model boundary conditions or climate-relevant constraints like ECS. Given such, I cannot in good faith recommend this manuscript for publication. The overarching concerns I have are that 1) the reconstruction decisions being made seem almost entirely *ad hoc* and otherwise unsupported in the present context and, to this end, 2) no validation efforts have apparently been attempted by the authors. From a paleoclimate- and proxy-interpretative standpoint, the study's scientific explorations feel rather cursory (e.g., the Discussion comprises only a paragraph or two of text), and I do not feel Annan et al. clearly answer their manuscript's own title, "Can we reliably reconstruct the mid-Pliocene Warm Period with sparse data and uncertain models?" I elaborate:

1. As a statistical methodology, offline data assimilation permits one to turn various "knobs" when generating paleoclimate reconstructions. While to first order this includes decisions on *which* models and proxies to include in the assimilation (aspects I am generally fine with in the authors' study, though please see Specific Point c), below), other important knobs include the degree of covariance localisation applied and the amount of uncertainty attributed to the individual proxy observations. Despite having considerably less proxy constraints as well as different models, Annan et al. are largely content to simply follow the empirical methodology presented in AHM22 without providing supporting evidence (such as careful validation testing; see Point 2, below) relevant to the present datasets / context. Further, Annan et al. include several pre-processing steps in their assimilation approach that do not appear supported in this study or in the prior paleoclimate literature outside of AHM22. I elaborate:

  i.    The authors apply a localisation of 2500km as in AHM22. While I am highly sceptical of such a low value in general (based on prior validation efforts using similar data/methods, e.g., Tardiff et al., 2019, Tierney et al., 2020, Osman et al., 2021, , Erb et al., 2022 etc., each of whom use localisation ≥12,000 to 24,000 km for significantly larger proxy compilations), I am willing to accept its use in AHM22 where some, if limited, validation efforts were performed and where proxy data coverage was well over an order-of-magnitude larger. This is not the case in the present study, wherein only a *maximum* of 23 locations are assimilated (and only 14 in the "preferred" set-up), and thus the majority of points on Earth (given regional clustering of those proxies that do exist) never even "feel" an update in the assimilation process. Indeed, I suspect this small localisation radius is likely why the authors feel compelled to bias-adjust ("re-center", see Point *iii*. below) their multi-model ensemble mean to the proxies in the first place, given all grid cells outside the collective proxies' 2500km radii will simply remain at the proxy-adjusted multi-model ensemble mean state. See also Specific Point e), below.

  ii.   Second, given the core importance of error quantification in data assimilation, the authors are rather *ad hoc* in their treatment of proxy uncertainties in assuming a constant 2°C error (1$\sigma$) across their proxies. While I understand this is an approach the authors have attempted to justify in their past work (again, AHM22) and even later in the present manuscript (see Specific Point f), below), I would appreciate greater effort be taken here in exploring the

influence of proxy uncertainties across sites and proxy types. Prior authors (e.g., Tardif et al., 2019, Tierney et al., 2020, Erb et al., 2021) have gone through considerable effort to validate proxy uncertainties and test their sensitivity. Further, modern proxy system models have been designed specifically to attempt to formalise (in the Bayesian sense, albeit still using empirical relationships) the magnitude-dependent uncertainties assigned to alkenones and Mg/Ca either in temperature or proxy space, yet the authors curiously disregard use of such tools (e.g., page 5, lines 10-13).

iii. Third, the authors invoke various pre-assimilation steps involving EOF's truncations and bias-adjustments of their multi-model mean. It is noted, however, that this re-centering step is not used nor recommended by the vast majority of paleo-data assimilation practitioners, given that it can precondition the posterior result significantly (c.f., the authors' Fig. 3a and Fig. 3b vs. 3c), and thus potentially bias the answer *further* from the "true" state which is unknown. While I can imagine such re-centering may be permissible in certain instances in a modern context where the true state is more or less understood (e.g., operational weather forecasting), who's to say that proxies are perfectly reliable (non-biased) representations in the paleo world? Indeed, this paper's subsequent claims that many of their proxies are *not* reliable indicators of the mPWP (Pg. 7, L34-35) implies a logical inconsistency with their re-centering step more generally.

Thus, the authors' pre-assimilation steps should be either supported by mathematical derivation (not provided in AHM22), reference to the prior relevant literature (also not provided in AHM22*), or, at minimum, empirical validation testing (provided in some capacity in AHM22). Furthermore, the authors' suggestion that rank histograms (Fig. 2, Pg. 6, L15-25) provide such missing validation to support their re-centering step is flawed: discovering a more uniform rank histogram (indicating that roughly half the proxy-derived temperatures sit below the re-centered median PlioMIP temperature, half above) after re-centering the model priors around the proxies is nothing more than the expected outcome of the same re-centering process! And, there are other caveats to the rank histograms still: for example, whereas prior approaches (e.g., Tierney et al., 2020) have tested rank histograms using withheld validation data only – a much more challenging test – this is almost certainly not the case here given the authors' lack of validation testing. In fact, it's unclear how these rank histograms were even calculated. Overall, if the authors wish to use their re-centering step, then use this technique should be subject to stringent validation testing using withheld or independent proxy constraints, mathematical underpinning, and (or) reference to the prior relevant literature showcasing its utility*.

*In introducing their re-centering step in AHM22, the authors cite only their previous study, Annan and Hargreaves (2013), which does not entail a data assimilation-based reconstruction approach and thus does not appear to be a relevant citation in the present context.

2. The authors do not validate their assimilated results using withheld or independent data. I often tell colleagues: "Anyone can create a data assimilation reconstruction (given the various toolsets and data compilations now openly available to do so) but not every reconstruction can be reliably validated." To me, this is what Annan et al. have provided: an interesting data analysis, to be sure, but without validating their reconstructions it's impossible to gauge which (if any) of the authors' various assimilated results should be believed, or whether any of their results actually improve upon the models or proxies in isolation? There are numerous approaches to validating a paleoclimate data assimilated result – the least stringent I'd hazard being leave-one-out proxy validation, the more stringent options being random sample without replacement, regional proxy withholding, or validation using independent (e.g., terrestrial) proxies. When rationalizing each of the various assimilation decisions and "knobs" noted above, the authors should be showcasing targeted validation efforts that support their claims.

Specific Points (Authors' text in quoted italics)

a) Pg. 5 L27-29 "*The Mg/Ca might be considered less reliable as their relationship to SST may depend on site-specific factors including the species analysed, the calibration used, or the seasonality or depth*

*habitat of the foraminifera.*" And, subsequently, Pg. 8 L16-18 "*...we (argue) that the Mg/Ca may be less reliable. We do not explore possible reasons for this here, but they may include site-specific factors including the species analysed, the calibration used, or the seasonality or depth habitat of the foraminifera (e.g. McClymont and Ho et al., 2023).*"

Yet, this is exactly what I feel this manuscript *should* be exploring. Several tools, imperfect or not, *do* exist to facilitate these aims (e.g., Tierney et al., 2019, Gray and Evans, 2019; see also review by Rosenthal et al., 2022). Indeed, understanding the sensitivity of proxy systems during past warm intervals is, to me, a key research area where paleoclimate data assimilation stands to permit real intellectual gains. Past efforts (Tardif et al., 2019, Tierney et al., 2020, Osman et al., 2021) have chosen to assimilate in proxy units precisely for the reason that doing so permits possibility of exploring sensitivity of the assimilation results to, e.g., underlying seasonal bias, depth-habitat influences, species sensitivities, pH, carbon dissolution effects, and sea surface salinity (SSS; among others). Assuming Mg/Ca reflects annual temperatures does not permit such explorations, unfortunately. Taking one example, recent results have suggested a much stronger change in SSS during the past several Ma than previously recognised, which could have influenced Mg/Ca-derived SST during the mPWP; this was not considered in the PlioVar data used here (Rosenthal et al., 2022). Finally, as a minor note, the authors' decision to omit BAYMAG-derived SST-estimates seems unfounded (Pg. 5, L10-13) especially given their decision to average alkenone-derived SST based on two separate estimates.

Speaking of …

b) Pg 5, L4-5. "*The UK37 SST values we take here are the simple average of the calibration of Müller et al. (1998), and the BAYSPLINE calculation, as presented in McClymont et al. (2020).*"

Averaging Muller and Tierney alkenone-derived SST values will be problematic for the tropics, where Tierney and Tingley (2018) illustrated a clear non-linearity of UK37 to SST as UK37 values approach their limit at 1. As this was not accounted for in the Muller et al. (1998) relations, Tierney should not be averaged with Muller-derived SST values in the tropics. (And, given the separation that will be required there, it would be more preferrable that alkenone-derived SST estimates be separated everywhere else.)

c) Pg 2, L31-34 "*While this has the unfortunate effect of reducing the number of usable data points, we note that the points that are masked in this way are coastal in location, which are potentially problematic for data-model comparisons due to the local nature of upwelling dynamics that is not always adequately captured by models.*"
And, subsequently, Pg 4 L22-24 "*This PRISM4 compilation contains 37 data points, reducing to 34 distinct grid points on the regular 5 × 5 degree grid that we use for our SST analysis, of which 23 locations remain after masking to the ocean grid of our ensemble.*"

Not all coastal regions are necessarily areas of downwelling, and in an interval as data sparse as the mPWP it would be useful to know where data are being lost and what effect they would have had on the assimilation should they have been kept. It is concerning that the authors are losing nearly 40% of their potential constraints due to their coarse re-gridding procedure alone. I'd ask that the authors explore this assumption further by either not masking their 5x5° grids that intersect land or, alternatively, assimilating these near-coastal points against the nearest available SST value.

d) Pg 6, L24-25 "*We then adopt this recentred ensemble as a prior for a standard Ensemble Kalman Filter (EnKF) assimilation step using the proxy data, similar to that of AHM22 and Tierney et al. (2020).*"

Tierney et al. (2020) did not apply this approach.

e) Pg 7, L18-23 "*If we do not perform the recentering and instead just perform the single step standard EnKF update to the original PlioMIP ensemble, the update is slightly greater than the EnKF step in the*

*two-step method, and generally more positive, due to the data being warmer than the ensemble mean. However the increment is still small, as the data are sparse and uncertain. Large areas of the globe are almost unaffected by the assimilation, with a temperature change of less than 0.1˚C. This is an inevitable consequence of having limited sparse data, and points to the influence of the model prior on the final result. Thus, with so few and uncertain data points the final result using a one-step framework would be very heavily based on the initial ensemble.*"

I find most of this text framed in a very misleading way. First, the fact that large areas of the globe are largely unaffected by the assimilation is not a surprise at all: it's an inevitable consequence of the fact that the assimilation is based on a mere 14 to 23 regionally clustered data points, each with a rather small 2500km localization radius. As noted earlier, this means that a substantial portion of the globe isn't even being updated in the assimilation. In fact, if the color scheme used by the authors in Fig. 3c were centered with, for example, a white color value atop $\Delta T = 0˚C$, I'd wager most of the globe would show in white rather than (the somewhat misleading) red or blue it currently is. Second, the authors' suggestion that their two-step process (involving an initial re-centering of their multi-model mean atop the proxy estimates prior to assimilation) is somehow "improving" upon the one-step process (assimilation only) is similarly misleading. Given that updates in the two-step process are even smaller than those without it (as expected given the reduced model-proxy offset, c.f., Fig. 3b), the two-step data assimilation could reasonably be viewed as being *less* of an improvement over the one-step method in the absence of validation efforts, since the two-step posterior remains closer to its prior state.

f) Pg. 7, L28-33 "*We take a uniform uncertainty of 2◦C on all of our data points. With so few data points, this estimate is necessarily itself uncertain, but we consider it reasonable for the following arguments. The RMS difference between the original PlioMIP ensemble members and the data points is around 2.6˚C averaged across the ensemble members, or 2.2˚C relative to the ensemble mean, which precludes a much higher error value since the data should not be closer to the models than they are to reality under the assumption that model errors and data errors are independent. Conversely, our posterior mean estimate after fitting to the data only achieves a residual RMS difference of 1.7˚C.*"

First, rationalizing the choice of proxy uncertainty by comparing proxy-inferred offsets to the same models you wish to assimilate appears to be a circular argument, since the true temperature state is unknown. Second, the RMS difference of the Mg/Ca-inferred temperatures to the posterior is also a meaningless comparison, given that any updated RMS difference will simply be an uncertainty-weighted reflection of the same Mg/Ca data that went into the assimilation! Indeed, had you increased the uncertainty to infinity, the RMS difference would remain 2.2˚C; for increasingly small uncertainty, the RMS difference would converge toward 0˚C. Please see Point 1ii., above.

g) Pg. 8, L8, L1-4 "*We adopt the same parameters for the algorithm that were shown to work well in AHM22, of 4 EOFs, and a localisation length scale of 2500km. While changing these values altered the regional patterns somewhat (for example, using a larger number of EOFs introduced more spatial variability) they made little difference to the large scale results such as global mean temperature anomalies.*"

The authors should either illustrate this visually or via statistical validation testing.

h) *Table 1*. More information on each model would be appreciated. At minimum, the degree of mPWP warming / cooling (SAT's and SST's) relative to their PI-reference would be useful to reference for each model.

**Citations**:

Annan, J. D. and Hargreaves, J. C. A new global reconstruction of temperature changes at the Last Glacial Maximum, *Clim. Past*, 9, 367–376, 2013. https://doi.org/10.5194/cp-9-367-2013

Annan, J. D., Hargreaves, J. C., and Mauritsen, T. A new global surface temperature reconstruction for the Last Glacial Maximum, *Clim. Past*, 18, 1883–1896 (2022). https://doi.org/10.5194/cp-18-1883-2022

Erb, M. P., McKay, N. P., Steiger, N., Dee, S., Hancock, C., Ivanovic, R. F., Gregoire, L. J., and Valdes, P.: Reconstructing Holocene temperatures in time and space using paleoclimate data assimilation, *Clim. Past*, 18, 2599–2629 (2022). https://doi.org/10.5194/cp-18-2599-2022

Gray, W. R., & Evans, D. Nonthermal influences on Mg/Ca in planktonic foraminifera: A review of culture studies and application to the last glacial maximum. *Paleoceanography and Paleoclimatology*, 34, 306–315 (2019). https://doi.org/10.1029/2018PA003517

Osman, M.B., Tierney, J.E., Zhu, J. et al. Globally resolved surface temperatures since the Last Glacial Maximum. *Nature* 599, 239–244 (2021). https://doi.org/10.1038/s41586-021-03984-4

Rosenthal, Y., Bova, S., & Zhou, X. A user guide for choosing planktic foraminiferal Mg/Ca-temperature calibrations. *Paleoceanography and Paleoclimatology*, 37, e2022PA004413 (2022). https://doi.org/10.1029/2022PA004413

Tardif, R., Hakim, G. J., Perkins, W. A., Horlick, K. A., Erb, M. P., Emile-Geay, J., Anderson, D. M., Steig, E. J., and Noone, D. Last Millennium Reanalysis with an expanded proxy database and seasonal proxy modeling, *Clim. Past*, 15, 1251–1273 (2019). https://doi.org/10.5194/cp-15-1251-2019

Tierney, J.E., Zhu, J., King, J. et al. Glacial cooling and climate sensitivity revisited. *Nature* 584, 569–573 (2020). https://doi.org/10.1038/s41586-020-2617-x

Tierney, J. E., Malevich, S. B., Gray, W., Vetter, L., & Thirumalai, K. Bayesian calibration of the Mg/Ca paleothermometer in planktic foraminifera. *Paleoceanography and Paleoclimatology*, 34, 2005–2030 (2019). https://doi.org/10.1029/2019PA003744

Tierney, J. E., & Tingley, M. P. BAYSPLINE: A new calibration for the alkenone paleothermometer. *Paleoceanography and Paleoclimatology*, 33, 281–301 (2018). https://doi.org/10.1002/2017PA003201

---

## Author Comment (AC1)

Reply to Reviewer 2. Review is (selectively) quoted in italics, with our responses interleaved

Thank you for the detailed comments. We agree with some points, but unfortunately the reviewer appears to have misunderstood some aspects of our work, which may indicate a lack of clarity in our manuscript. We will attempt to improve the explanations throughout the manuscript. We address the reviewer's comments in the order presented.

*[...]However, numbers like those mentioned above matter — they get passed on in the literature for generating model boundary conditions[...]*

-> We believe we have included sufficient caveats about the results such that we cannot reasonably be blamed in advance for the hypothetical argument that others may use our results incorrectly. To quote from our conclusions:

"...the different data sets produce rather different estimates ranging from 1.0 to 4.7∘C for the best estimate of global surface air temperature anomaly. All the data sets are sparse with high uncertainty, and therefore our confidence in our result is not very high. We think that the regional scale information in the reconstruction is not likely to be reliable..."

"With such small data sets as we have here, the models also necessarily play an uncomfortably large role. We have investigated the effect of using the PlioMIP models themselves as a prior, versus recentering the ensemble on the data. This choice has a significant influence on the results. While in principle we prefer the data-centred approach, this is not an unquestionable choice to make."

If the reviewer has any specific concerns about the wording in our manuscript, it would be helpful for them to suggest ways that this could be made clearer.

*1 [..] Despite having considerably less proxy constraints as well as different models, Annan et al. are largely content to simply follow the empirical methodology presented in AHM22 without providing supporting evidence (such as careful validation testing; see Point 2, below) relevant to the present datasets / context. [...]*

-> It is precisely because of the small data set that such validation is not viable. We will explain this more clearly in the revised manuscript. With a prior predictive RMS error of about 2.8 degrees and only a dozen points, the standard error on this estimate is itself around 0.8C (implying a 95% confidence interval of plus or minus twice this value). We have already shown in AHM22 that the reconstruction is not very sensitive to parameter values

used in the method, and in that work we had 400 data points such that more modest differences in outcome could potentially be identified. Furthermore, we have already shown here that the reconstruction is highly sensitive both to the choice of data set and also to the choice of recentering (or not) the model prior. It is hard to imagine that the small changes that could arise from any reasonable changes in parameter values could alter these conclusions. We did of course perform a larger number of sensitivity tests than are presented in the manuscript but the results of these were unremarkable, as expected following those presented in AHM22, and were omitted to keep the manuscript readable. Some will be added as mentioned later in this reply

*i. The authors apply a localisation of 2500km as in AHM22.*

-> This comment appears to be based on a misunderstanding by the reviewer. The 2500m value is the half-width of the localisation cut-off, as was made clear in AHM22 (and also in the code). Specifically, it's the parameter "c" in the Gaspari and Cohn formulation. Thus, the reviewer's assertions on this point (and also Specific Point e later) are incorrect, as the cut-off is 5000km which while smaller than Tierney et al, still avoids there being large data voids. We will revise the text to include the explanation from AHM22, in order to avoid other readers making this mistake. The recentering process already ensures that each data point has global influence regardless of localisation in the EnKF. However in the case where recentering is not performed, we agree it would be reasonable to use a greater localisation length scale. We will therefore present the results of a test using a greater localisation length scale together with the uncentred approach.

*ii Second, given the core importance of error quantification in data assimilation, the authors are rather ad hoc in their treatment of proxy uncertainties in assuming a constant $2^{\circ}C$ error ($1\sigma$) across their proxies.*

-> It is not within the scope of this work or the expertise of the primary authors to explore in detail the modelling of uncertainties in proxy analysis. We also think that it is obvious that the large uncertainty in our result, dominated as it is by the choice of data and ensemble recentering methodology, will not be significantly altered by such second-order effects as the detailed modelling of uncertainties of these data points. Code is of course available for any other researchers who wish to perform such investigations, and we believe that other researchers may be better placed than us to explore these issues.

*iii Third, the authors invoke various pre-assimilation steps involving EOF's truncations and bias-adjustments of their multi-model mean. It is noted, however, that this re-centering step is not used nor recommended by the vast majority of paleo-data assimilation practitioners, given that it can precondition*

*the posterior result significantly (c.f., the authors' Fig. 3a and Fig. 3b vs. 3c), and thus potentially bias the answer further from the "true" state which is unknown.*

-> Thank you for noting the originality of our work. There has indeed been very little discussion of the importance of the prior in paleoclimate reconstructions of this type, a regrettable state of affairs that the authors accept some blame for. Indeed our first work in this area (Annan et al, Scientific On-Line Letters on the Atmosphere, 2005) used a single model with varying parameter values in an attempt to represent uncertainties in climate feedbacks. It was only subsequent to this work that we came to more clearly understand the limitations of such an approach. While we have subsequently shown that multi-model ensemble provides a more robust approach, we also demonstrated in AHM22 that any significant biases in such a multi-model prior would pass through in the posterior, even in that scenario where we had 400 data points distributed widely over land and sea. See Sections 5.2 of AHM22 for analysis and discussion of this issue, and also Section 6 of that paper for further comparison with Tierney et al 2020. With only at most two dozen data points, it is inevitable that the choice of prior is a critical factor in this current work and the use of an ensemble of convenience simply because it's what everyone else does is not tenable. We had hoped that this point was already well enough made in AHM22 but perhaps it bears repeating in this manuscript.

We note that the reviewer does not present any scientific arguments in favour of using the uncentred multi-model ensemble, let alone the single model ensembles that are still sometimes used in this area of research. Our method did not appear from a vacuum but rather through a critical analysis of previous work, including our own. We hope that other researchers working in this area will follow our lead in considering more carefully the sensitivity of their results to the priors that they use.

We already emphasise in the manuscript that the reconstruction is strongly dependent on the recentering decision. If other researchers have reason to prefer the uncentered ensemble, that option is available to them.

*2. The authors do not validate their assimilated results using withheld or independent data.*

-> We return to the point made previously, that with only a dozen or so data points (a maximum of 23), there is not any hope of meaningful validation of the approach in this application, which is why we rely on the validation performed in ANH22. We will explain this point in the revised manuscript. The EnKf itself is of course decades old and well established. We are not presenting further methodological innovations here, merely applying the

approach of AHM22 to a different time period.

Specific points:

*a Yet, this is exactly what I feel this manuscript should be exploring.*

-> However, this is not the area of expertise of the primary authors. As in our replies to reviewer 1, we can highlight some of the reasons for these discrepancies in a revised manuscript but this remains a highly active area of research. A recent review showed that the absence of multi-proxy single-site analyses has significant impact on addressing this issue (McClymont & Ho et al., 2023). We are aware of ongoing work which is specifically seeking to understand the apparent cold bias in Mg/Ca Pliocene temperatures but this work is early in its development and not available for discussion here. Many of the parameters which could explain this bias (e.g. salinity, seawater chemistry, carbonate dissolution, seasonality, depth habitat) are even less well constrained for the Pliocene, and in many cases can't yet be quantified (McClymont & Ho et al., 2023). We re-emphasise here that our focus is to explore the impact of proxy choice and site distributions on data-model assimilation, and that by showing this impact we hope that this will motivate further work to investigate why.

b

-> The same comment as for (a) above applies.

We agree with the reviewer that by by averaging the two alkenone data sets we do intrinsically reduce the low latitude error (in that BAYSPLINE is more like a 4*C uncertainty) whereas in the high latitudes we're creating a value which sits between two calibrations, even though the difference between them is <0.5*C (with an overall calibration error more like 1.5*C). As per our reply to the previous comment, we can highlight this issue in more detail and explain the rationale for the omission of BAYMAG data.

c

-> The suggestion of ad-hoc movement of data points does not seem entirely consistent with the reviewer's complaints regarding a number of decisions we've already taken. Where models cannot resolve coastal areas adequately, model-data comparison is always going to be challenging. Furthermore, some of the omitted data points are those in the Benguela upwelling area where where there are significant issues.

d

-> We will change the wording

e

-> The 2500km issue rears its head again. The reviewer's comments about a substantial part of the globe being unaffected is incorrect. We will present a test using a longer length scale to demonstrate that this issue does not affect our conclusions.

f.

-> The argument is not circular when we perform the comparison to the model prior. The argument is based on the simple observation that the data errors cannot plausibly be greater than the (prior) model-data difference, since models also have errors when compared to the unknown truth, and the modelling errors can be reasonably assumed independent of data errors. Similarly, pairwise model differences provide some evidence as to the magnitude of model errors, though this can only ever indicate a lower bound on such errors, as we cannot reasonably assume model errors are independent across the model ensemble. Of course sampling errors also limit the precision of what can be reasonably concluded from these analyses, but they are still relevant information.

g

-> We will add pictures to the supplementary information

h

-> We will add the values to this table.

---

## Author Comment (AC2)

Reply to Reviewer 1. Review is quoted in italics, with our responses interleaved.

*The manuscript by An[n]an et al. is a useful addition to the literature, clearly laying out the issues of model uncertainty and the relatively sparse data available to compare to those models. The methodology used has been used before, by the authors, to reconstruct LGM conditions. Here they apply the same methodology to the mid Piacenzian (Pliocene).*

-> Thank you for the comments.

*The paper is a useful example of the methodology and poses questions that should be followed up by those looking at Pliocene and other deep-time climates. There are a number of minor issues, enumerated below, that detract from the overall message. If one is going to compare different proxy data sets, an attempt should be made to use as close to apples vs apples as one can get. A comparison of the PlioVAR and PRISM allkenone compilations, which use basically the same data, would be more informative if the same resolution was chosen. Instead of comparing the data from ± 10 kya windows around 3.205 Ma, a comparison was made using ±10 kya for one data set and ±15 kya for the other. This may not make much difference but could have been avoided.*

-> These data sets have been made available to the modelling community as products for comparison with and validation of models. Our aim in selecting them was to investigate their implications for reconstruction of global temperature fields, rather than for the purposes of direct comparison. The windows are different because this is the nature of the data sets that have been provided to the modelling community (e.g. Haywood et al., 2020 uses the ±15 ky window of Foley and Dowsett for PlioMIP2 data-model comparison, whereas McClymont et al., 2020 focus on the ±10 ky window). Although the Foley and Dowsett database made available a 10K window which aligns with PlioVAR, the stratigraphic age controls on the sites used by PlioVAR were also reviewed and, in some cases, revised (outlined in McClymont et al., 2020) which also introduces differences between the two data sets. As the reviewer notes in their next comment, the impact of choosing the different data sets is minor, so we prefer not to have a detailed discussion about possible influences on differences which could detract from our main message.

*Figure 4 shows anomaly maps for (a) PRISM4, (b) PlioVAR Mg/Ca and (c) PlioVAR ALL. It would be helpful to see a plot of PlioVAR Uk37 for comparison (I think this is Figure 1 (b)). Having it side by side as part of Figure 4 would make visual comparison of the different data sets more productive. The*

*differences between PlioVAR Uk37 and PRISM4 are minor, and both show marked differences compared to PlioVAR Mg/Ca. This isn't a surprise and is nicely documented quantitatively, but seeing adjacent images would help.*

-> We will duplicate the plot from Fig 1 in Fig 4 if editorial staff allow.

*The conclusion that the models may be underestimating polar amplification isn't much of a surprise to the community, but it is useful to document it as the authors have. Likewise, much is made of the mismatch between Mg/Ca and alkenone based SST estimates. This is nothing new, having been discussed in more detail in countless previous papers.*

-> We agree with your comments about polar amplification.
As for the data mismatch, yes we agree this is not new but hope it is useful to emphasise the importance of this issue. The use of model fields to interpolate between sparse data points allows for a more comprehensive comparison across sites that are not co-located, when compared to a direct comparison of data points alone.

Thank you also for pointing out various mistakes in the references, which we will tidy up.

*page 4 line 2: This appears to be a simple misunderstanding, but Bragg (2014) could not have used PRISM4 data since those data were not available prior to 2016 and SST estimates shown in Foley and Dowsett (referred to here as PRISM4), were not produced until 2019.*

-> Yes, this will be corrected. Bragg used the PRISM2 and PRISM3 "time slabs" rather than the single interglacial KM5c (PRISM4).

*Page 4, lines 18-19: Why use the PRISM4 community sourced verification data with a 30K window to compare to PLIOVAR's 20K window when PRISM also, in the same release, produced a version with a ±10K window?*

-> In PlioMIP2 data-model comparisons (e.g. Haywood et al., 2020) which use Foley and Dowsett, 2019) the data set is referred to as the "PRISM4 SST data". In the original definition of this data set (Figure 1, Dowsett et al., 2016) the PRISM4 time series is defined as encompassing Marine Isotope Stage M2 through to interglacial KM3 (3.190 to 3.220 Ma) which is the 30 kyr we have used in the manuscript.

*Page 4, lines 28-29: The PlioVAR interval is slightly narrower only due to your choice of the 30 kyr window rather than the identical 20 kyr window provided in Foley and Dowsett (2019).*

-> please see our reply to the previous question.

*Page 5, lines 26-27: You should probably cite a couple of the many available references that previously documented differences between Mg/Ca and alkenone based SST estimates in Pliocene and Pleistocene sequences.*

-> Agreed, we can add a line here to say "as observed in some of the original time series" and provide references to support the statement.

*Page 5, lines 30-34: This is an interesting point and it would be helpful if it was addressed in this paper. Foley and Dowsett (2019) is a compilation of previously published alkenone data and it would be useful to know whether the sites not in common with PlioVAR are from a particular region, particular lab, etc.*

We can provide a note to this effect in our revised manuscript. Broadly speaking, the sites which were not included in PlioVAR were the result of not meeting the PlioVAR time resolution constraints required for the data and/or the age model, and tend to be from studies where the focus was on low-resolution analysis of longer time series rather than any regional or lab specific bias.

*Page 7, lines 33-35: As in one of the comments above, comparing alkenone and Mg/Ca based SST estimates is like apples and oranges. They are measuring different things and while both are calibrated to mean annual SST, the literature is ripe with examples of the two providing discordant estimates. On page 8 of this manuscript you indicate some possible reasons for Mg/Ca data being less reliable, the same reasons that have been stated by many authors in the past. Maybe move those up to page 7 and provide citations to earlier works?*

Yes we can move some of the reasons up to an earlier part of the manuscript and provide some of the citations to earlier works. A challenge is that this is a complex subject which has also been reviewed and discussed extensively in other papers, including for the reasons given by the reviewer. We will ensure that relevant citations can direct readers to this issue. Our focus here was to see what impact including "all" or "selected" proxy data had on the model results rather than explaining proxy differences.

---

## Author Response (AR1)

Final response to reviewer 1

Regarding the main comment relating to the choice of 30ka interval for the PRISM4 data set, we now additionally mention results generated from the tighter 10ka PRISM4 interval. However, we also modified our wording to clarify that these values are the full widths of the two PRISM4 intervals, ie they are defined by the ranges 3.205+-0.015Ma and 3.205+-0.005Ma respectively, and there is no PRISM4 data set using an interval of 3.205+-0.01Ma that would be directly equivalent to that of McClymont et al. In any case, the two versions of the PRISM4 sets are very similar and generate near-identical results.

Individual notes:

*Page 4, line 2:*

*This appears to be a simple misunderstanding, but Bragg (2014) could not have used PRISM4 data since those data were not available prior to 2016 and SST estimates shown in Foley and Dowsett (referred to here as PRISM4), were not produced until 2019.*

The reference made to the data set used by Bragg has been corrected.

*Page 4, lines 18-19:*

*Why use the PRISM4 community sourced verification data with a 30K window to compare to PLIOVAR's 20K window when PRISM also, in the same release, produced a version with a ±10K window?*

*Page 4, lines 28-29:*

*The PlioVAR interval is slightly narrower only due to your choice of the 30 kyr window rather than the identical 20 kyr window provided in Foley and Dowsett (2019).*

As above, neither of the PRISM4 intervals coincides with that of PlioVAR. We have tested both versions of PRISM4 and report this in the paper.

*Page 5, lines 26-27:*

*You should probably cite a couple of the many available references that previously documented differences between Mg/Ca and alkenone based SST estimates in Pliocene and Pleistocene sequences.*

Additional comments and references added

*Page 5, lines 30-34:*

*This is an interesting point and it would be helpful if it was addressed in this paper. Foley and Dowsett (2019) is a compilation of previously published alkenone data and it would be useful to know whether the sites not in common with PlioVAR are from a particular region, particular lab, etc.*

*Page 7, lines 33-35:*

*As in one of the comments above, comparing alkenone and Mg/Ca based SST estimates is like apples and oranges. They are measuring different things and while both are calibrated to mean annual SST, the literature is ripe with examples of the two providing discordant estimates. On page 8 of this manuscript you indicate some possible reasons for Mg/Ca data being less reliable, the same reasons that have been stated by many authors in the past. Maybe move those up to page 7 and provide citations to earlier works?*

Comments moved as requested

*Page 9, line 17:*

*I think you just made a simple typo with the citations. If you are referring to the PRISM4 compilation you must mean Haywood et al. 2020 (not 2010).*

This has been corrected.

Thank you also for the various corrections to references including DOIs.

Final response to reviewer 2.

Where we have disagreed with your suggestions in our previous response, we have not changed the paper.

In regards to your point about cutoff radius, we clarify that the cutoff radius is actually double the length parameter that we've quoted. we now make specific reference to the tests we performed with a greater length parameter of 10,000km which demonstrate that this is not a major factor in our results regardless of recentering the prior.

We have also emphasised more strongly that the prior is inevitably a

significant factor in the results, and added a reference to a newly published LGM/deglaciation reconstruction which further demonstrates this point. Reconstructions based on a single model ensemble cannot reasonably be considered robust.

We have explained that leave one out cross validation isn't a meaningful exercise in this situation due to the extremely low number of data points. However we also observe that the tests of various parameters in the reconstruction process have relatively little influence especially in contrast to the choice of data set and recentering (or not) the model prior.

We do include the global surface air temperature anomaly for each model, in Table 1.

---

## Author Response (AR2)

[revised manuscript text omitted]

(a) Recentering step of two-step process

(b) EnKF step of two-step process.

[Figure]

(c) EnKF step of one-step process.

**Figure 3.** Comparison of two-step (recentering and Enkf) versus single step (EnKF only) process. Figures (a) and (b) show the increments generated in the recentering step and EnKF step respectively in the two-step process using PlioVAR-$U_{37}^{K'}$. Figure (c) shows the increment generated in a single step EnKF procedure. Note the different scale on the colour bar for the EnKF steps

[Figure]

(a) Posterior means using PRISM4 data

(b) Posterior means, using PlioVar-Mg/Ca data.

[Figure]

(c) Posterior means, using PlioVar-all data.

**Figure 4.** Results using three different data sets.